# Latent Chain-of-Thought for Visual Reasoning

**Guohao Sun[1,2,∗] Hang Hua[2,3,+], Jian Wang[2,+], Jiebo Luo[3],**
**Sohail Dianat[1], Majid Rabbani[1], Raghuveer Rao[4], and Zhiqiang Tao[1]**

[1]Rochester Institute of Technology, [2]Snap Inc.,
[3]University of Rochester, [4]DEVCOM Army Research Laboratory

## Abstract

Chain-of-thought (CoT) reasoning is critical for improving the interpretability and reliability of Large Vision-Language Models (LVLMs). However, existing training algorithms such as SFT, PPO, and GRPO may not generalize well across unseen reasoning tasks and heavily rely on a biased reward model. To address this challenge, we reformulate reasoning in LVLMs as posterior inference and propose a scalable training algorithm based on amortized variational inference. By leveraging diversity-seeking reinforcement learning algorithms, we introduce a novel sparse reward function for token-level learning signals that encourage diverse, high-likelihood latent CoT, overcoming deterministic sampling limitations and avoiding reward hacking. Additionally, we implement a Bayesian inference-scaling strategy that replaces costly Best-of-N and Beam Search with a marginal likelihood to efficiently rank optimal rationales and answers. We empirically demonstrate that the proposed method enhances the state-of-the-art LVLMs on seven reasoning benchmarks, in terms of effectiveness, generalization, and interpretability. The code is available at `https://github.com/heliossun/LaCoT`.

## 1 Introduction

Chain-of-thought (CoT) reasoning is critical for enhancing the interpretability and reliability of Large Vision-Language Models (LVLMs) [11, 29, 7, 44, 15, 16]. These models combine visual perception and natural language processing to perform intricate reasoning tasks that require explicit, step-by-step rationalization. As LVLMs have expanded into more sophisticated applications, such as visual question answering, commonsense reasoning, and complex task execution, the limitations of current training methods, such as generalization, have become increasingly evident.

To enable visual CoT, mainstream training paradigms, such as Supervised Fine-Tuning (SFT), Proximal Policy Optimization (PPO) [34], and Group Relative Policy Optimization (GRPO) [12], primarily focus on optimizing next-token distributions or scalar rewards. While effective for in-distribution tasks, these methods often struggle to generalize across diverse reasoning questions due to their inability to explicitly capture dependencies across trajectories [14]. Specifically, SFT heavily depends on teacher-forced log-likelihood, only to parrot reference traces; meanwhile, PPO and GRPO are constrained in exploration as their KL penalties enforce proximity to the SFT baseline, making them fall short in finding novel rationales. Additionally, they may cause a reward hacking [36] issue that achieves high scores without genuinely solving the intended problem. To address these limitations, this work adopts a latent variable model to realize visual CoT as a probabilistic inference problem [6] over latent variables, allowing us to work with rich, expressive probabilistic models that better capture uncertainty and hidden structure, without needing direct supervision.

---

∗Part of this work was conducted while Guohao Sun and Hang Hua were interns at Snap Inc. +Hang Hua and Jian Wang contributed equally. Corresponding authors: Guohao Sun (gs4288@rit.edu) and Zhiqiang Tao (zhiqiang.tao@rit.edu)

Unlike using prompting and in-context learning to generate deterministic reasoning CoT ($Z$), we treat $Z$ as a latent variable sampled from a posterior $P(Z|X,Y) = P(XZY)/\sum_{Z'} P(XZ'Y)$, given a question-answer pair $(X,Y)$ as observation. However, such sampling is intractable due to the normalization term. Existing methods to sample approximately from an intractable posterior include Markov chain Monte Carlo (MCMC) and RL approaches such as PPO [34]. Despite good training efficiency, these methods show limited capacity in modeling the full diversity of the distribution [14]. By contrast, Amortized Variational Inference (AVI) [53, 19, 22, 23] yields token-level learning through optimizing the Evidence Lower Bound (ELBO), which encourages diverse trajectories and provides a principled way to draw samples from the target posterior distribution (see Fig. 1). One way to implement AVI is given by the generative flow networks (GFlowNets [4, 5]) algorithm: training a neural network to approximate a distribution of interest. Despite achieving strong performance in broad text reasoning tasks, prior GFlowNets-based approaches [14] have yet to fully address visual reasoning due to the long CoT sequence inherent in multimodal tasks(e.g., $\sim 1k$ tokens).

In this study, we propose a novel reasoning model, namely **LaCoT**, which enables amortized latent CoT sampling in LVLMs and generalizes across various visual reasoning tasks. To achieve this, we propose ❶ a general RL training algorithm (RGFN) with a novel reference-guided policy exploration method to overcome the catastrophic forgetting issue and eliminate the diversity constraint caused by the KL penalty. To improve exploration efficiency, we introduce ❷ a token-level reward approximation method, allowing efficient mini-batch exploration for diverse sampling. Finally, we introduce ❸ a Bayesian inference-scaling strategy (BiN) for optimal rationale-solution searching at inference time for any reasoning LVLM. Previous works have provided empirical evidence that Best-of-N (BoN) sampling [37], Beam Search [41], and other heuristic-driven approaches [47] can improve model's performance at inference time. However, these methods are computationally costly and rely heavily on biased critic models, failing to provide an optimal reasoning chain or answer efficiently. Our inference procedure is grounded in Bayesian sampling principles to eliminate the critic model and improve interpretability. We treat rationales as integration variables and rank answers by a principled, length-normalized marginal likelihood. Consequently, our method delivers a scalable, probabilistically justified searching strategy, effectively identifying optimal rationales and answers within LVLMs.

Empirically, we develop the proposed LaCoT on two base models, Qwen2.5-VL [3] 3B and 7B, where the 7B model achieves an improvement of $6.6\%$ over its base model and outperforms GRPO by $10.6\%$. The 3B model surpasses its base model with $13.9\%$ and achieves better results than larger models, e.g., LLaVA-CoT-11B and LLaVA-OV-7B, demonstrating the effectiveness of learning to sample latent CoT on reasoning benchmarks.

## 2 Preliminaries

Generative Flow Networks (GFlowNets) [5, 20, 4, 54] are a class of amortized variational inference methods designed to sample complex, structured objects such as sequences and graphs with probabilities proportional to a predefined, unnormalized reward function. Unlike traditional generative models that often focus on maximizing likelihood or expected reward, GFlowNets objective, such as Sub-Trajectory Balance (subTB) [27], is a hierarchical variational objective [28]. Such that if the model is capable of expressing any action distribution and the objective function is globally minimized, then the flow consistency for trajectory $\tau = (z_i \to \cdots z_j)$ is

$$F(z_i) \prod_{k=i+1}^{j} P_F(z_k \mid z_{k-1}) = F(z_j) \prod_{k=i+1}^{j} P_B(z_{k-1} \mid z_k) \qquad (1)$$

by minimizing a statistical divergence between the learned and and the target distributions over trajectories $D_{KL}(P_B||P_F)$, where $F(z_i)$ is the in flow at state $z_i$, $P_F(z_i|z_{i-1})$ and $P_B(z_{i-1}|z_i)$ indicates the forward and backward policy that predicts the probability between states.

In the case of causal LLM, token sequences are autoregressively generated one-by-one from left to right, so there is only one path to each state $z_i$, and each state has only one parent $z_{i-1}$. Given this condition, $P_B(-|-) = 1$ for all states. By modeling $P_F(-|-)$ with $q_\theta(-|-)$, parameterized by $\theta$, the loss function aims to ensure consistency between the flow assigned to all trajectories from one complete rationale (i.e., $Z = (z_1 z_2 \cdots z_n \top) = z_{1:n}\top$). Specifically, for trajectory truncated by a paired index $(i, j)$ with $0 \le i < j \le n$, the loss penalizes discrepancies between the flow at state $z_i$,

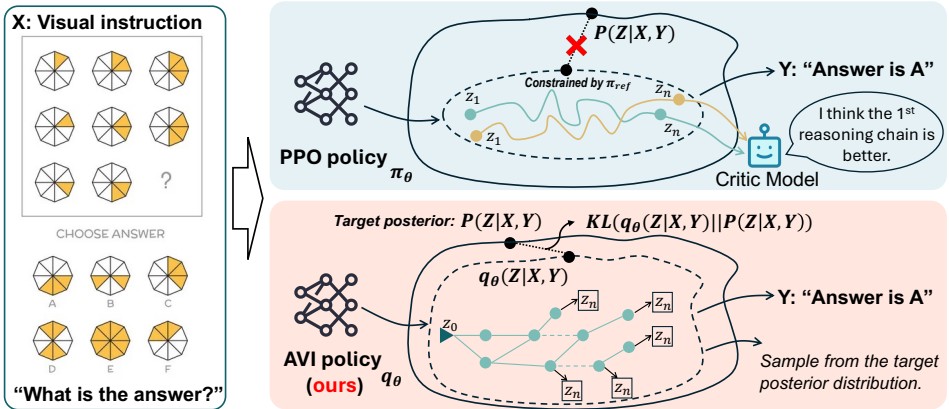

Figure 1: Comparison of different training algorithms for visual reasoning. PPO implicitly approximates the rationale distribution but tends to under-represent its full diversity due to limited exploration constrained by its reference policy (e.g., the SFT model), and it heavily relies on a critic (reward) model. In contrast, AVI explicitly estimates the true target posterior $P(Z|X,Y)$ through latent rationales, which promote diverse trajectories and inherently prevent reward hacking.

scaled by the product of transition probabilities from $z_{i+1}$ to $z_j$ and the flow at $z_j$:

$$
\begin{aligned}
\mathcal{L}_{\text{SubTB}}(Z;\theta) &= \sum_{0 \le i < j \le n} \left[ \log \frac{F(z_i) \prod_{k=i+1}^{j} q_\theta(z_k \mid z_{1:k-1})}{F(z_j)} \right]^2 \\
&= \sum_{0 \le i < j \le n} \left[ \log \frac{R(z_{1:i}\top) \prod_{k=i+1}^{j} q_\theta(z_k \mid z_{1:k-1}) q_\theta(\top \mid z_{1:j})}{R(z_{1:j}\top) \, q_\theta(\top \mid z_{1:i})} \right]^2,
\end{aligned}
\tag{2}
$$

where $F(z_i) = R(z_{1:i}) = \frac{R(z_{1:i}\top)}{q_\theta(\top|z_{1:i})}$ when $z_i$ is the the final state, $R(z_{1:i}\top)$ is the reward of trajectory ends at $z_i$, where $\top$ represents the terminal state, which is usually an $\langle eos \rangle$ token in LLM.

## 3 Amortizing Variational Inference for Latent Visual CoT

By leveraging GFlowNets for AVI in LVLM, we formulate visual reasoning as a variational inference problem, as shown in Fig. 1. That is, given a question-answer pair $(X, Y)$ as an observation, the goal is to find the latent visual CoT sequences $Z$ that contribute the most to the conditional likelihood:

$$
P(Y|X) = \sum_{Z \sim P(Z|X,Y)} P(ZY|X),
\tag{3}
$$

where $P(ZY|X)$ denotes the likelihood assigned to a concatenated sequence (e.g., $ZY$) given visual instruction $X$, and $Z$ is a latent CoT supposed to be sampled from a posterior distribution $P(Z|X,Y)$. To approximate such a posterior, we use GLowNets objective derived in Eq. (2) – an amortized variational inference method – to train an autoregressive model $q_\theta(Z|X)$. By minimizing Eq. (2), the policy model learns to generate trajectories where the probability of generating a particular trajectory is proportional to its reward (i.e., unnormalized posterior probability), ensuring that higher-likelihood rationales (as determined by $R$) are more likely.

However, ❶ Eq. (2) requires token-level reward, which is infeasible in complex reasoning chains with thousands of tokens. ❷ Efficient and diverse exploration remains a challenging research problem in reinforcement learning, especially when an environment contains large state spaces. Given these research problems, we provide our solutions in the following sections.

### 3.1 Token-level Marginal Reward Approximation

The proposed amortized rationale sampler $q_\theta(Z|X)$ shares the same generation process as in autoregressive LVLM: given a prefix condition $X$, and at the $i$-th step, a token $z_i$ is sampled from

a policy model $q_\theta(z_i|X, z_{1:i-1})$, which is then appended to the sequence. Consistent sampling autoregressively from the LVLM until a terminal state $\top$ is reached gives us one completion of rationale $Z = (z_1 z_2 \cdots z_n \top)$. As shown in Eq.(2), the objective function incorporates state-level rewards, enabling the model to correctly attribute the contribution of each step to the final reward. By setting the reward $R(z_{1:t}\top) = \log P(X z_{1:t} \top Y) \propto P(z_{1:t}\top | X, Y)$, we optimize the policy model to sample all trajectories such as $\tau = z_{1:t}\top$ from the target distribution at convergence.

By treating each token as a state, such a training algorithm provides clear guidance for the policy on how early actions impact the final outcome, helping reduce variance and improving convergence [27]. However, directly computing the exact reward for all states is computationally expensive during training, especially for a long rationale sequence. A natural approximation is to assume local smoothness of reward within small regions. To efficiently estimate intermediate rewards, we adopt a linear interpolation strategy within segmented regions of length $\lambda$ as shown in Fig. 2.

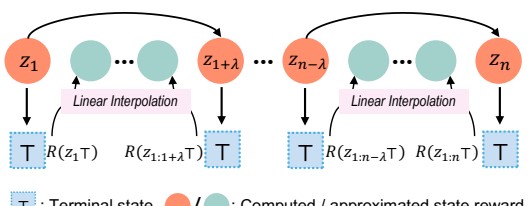

Figure 2: Within a complete rationale sequence, we compute the actual reward after each $\lambda$ steps and adopt a linear interpolation strategy to estimate the intermediate steps.

The following proposition summarizes our theoretical claim for improving the training efficiency of Eq. (2). This approximation leverages the local smoothness of the log-likelihood, significantly reducing computational overhead without substantial loss in accuracy. We empirically evaluate the effectiveness of our claim in the experimental section.

**Proposition 1.** *Let $R(z_{1:t}\top) = \log P(X z_{1:t} Y)$ be a joint-likelihood reward function.*

*(a) If $R(z_{1:-})$ and $R(z_{1:-+\lambda})$ are true reward and the intermediate rewards within region of length $\lambda$ are constantly increment, then we can approximate the reward at step $t + i$ (where $0 \leq i \leq \lambda$) as*

$$\tilde{R}(z_{1:t+i}\top) = R(z_{1:t}\top) + \frac{i}{\lambda}\left(R(z_{1:t+\lambda}\top) - R(z_{1:t}\top)\right). \tag{4}$$

*(b) If $\lambda$ is short enough, the interpolation reward error stays close to 0 and the flow between $F(z_{1:-})$ and $F(z_{1:-+\lambda})$ satisfies Eq. (1).*

*Proof.* See Appendix A. □

Substituting the estimated reward $\tilde{R}$ in Eq. (2) gives our modified interpolated sub-trajectory balance ($\mathcal{L}_{\text{ISubTB}}$) loss:

$$\mathcal{L}_{\text{ISubTB}}(Z;\theta) = \sum_{0 \leq i < j \leq n} \left[\log \frac{\tilde{R}(z_{1:i}\top) \prod_{k=i+1}^{j} q_\theta(z_k \mid z_{1:k-1}) q_\theta(\top \mid z_{1:j})}{\tilde{R}(z_{1:j}\top) \, q_\theta(\top \mid z_{1:i})}\right]^2, \tag{5}$$

where $\tilde{R}(z_{1:i}\top)$ is defined pice-wise as:

$$\tilde{R}(z_{1:i}\top) = \begin{cases} R(z_{1:i}\top) & \text{if } i \text{ is the index of actual reward,} \\ R(z_{1:t}\top) + \frac{i-t}{\lambda}\left(R(z_{1:t+\lambda}\top) - R(z_{1:t}\top)\right) & \text{if } t < i < t + \lambda \text{ (estimated)} . \end{cases}$$

By computing the sparse rewards and efficiently approximating the intermediate states' rewards, we can easily apply mini-batch exploration for diverse sampling to improve the generalizability of $q_\theta(Z|X)$ by covering the full target posterior.

## 3.2   Reference-Guided GFlowNet Fine-tuning

Previous works [13, 26] suggest that exploration can let policy gradient methods collect unbiased gradient samples, escape deceptive local optima, and produce policies that generalize better. However, as shown in Fig. 3, allowing the model to explore without constraint causes the catastrophic forgetting issue, where the model tends to generate meaningless content with high likelihood but low reward. Existing methods, such as KL penalty [33] and clipped surrogate objective [34], control the size

of the gradient update. If the resulting policy is too far from the previous policy, the KL penalty constrains it to take an overly aggressive learning step. However, such a method limits the exploration and increases the variance of trajectories [45]. To address this issue, we propose a simple but effective solution by integrating a reference-based mechanism to guide the exploration process towards generating higher-quality rationales.

During training, we first explore $m$ candidate latent rationales $\{Z_1, Z_2, \ldots, Z_m\}$ from the current policy model $q_\theta(Z|X)$ and compare them against a reference rationale $Z_{\text{ref}}$ that anchors the search in a data-grounded region. Before gradient descent, each candidate $Z_i$ is associated with a reward $R(Z_i) = \log P(XZ_iY)$, and the ones that underperform the reference rationale are discarded before they reach the gradient, preventing collapse into a meaningless but high-probability trajectory.

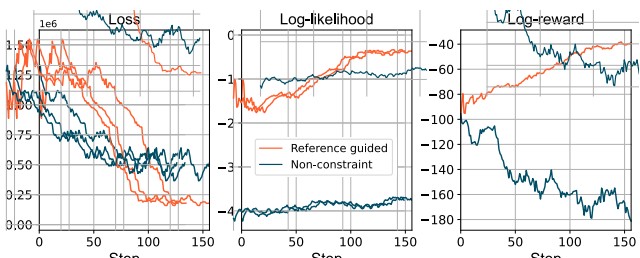

Figure 3: Allowing the policy model to explore the state space without constraint causes the catastrophic forgetting issue. The proposed reference-guided exploration effectively addresses this problem.

To achieve candidate filtering, we define an indicator function:

$$\mathbb{I}(Z_i) = \begin{cases} 1, & \text{if } R(Z_i) > \delta_s R(Z_{\text{ref}}) \\ 0, & \text{otherwise} \end{cases} \tag{6}$$

where $\delta_s = \tau_{max} - (\tau_{max} - \tau_{min}) * min(1, s/50)$ is the annealing coefficient, $s$ is the index of the current training step. By doing this, we tolerate more exploration at the beginning and gradually increase its standard. The acceptance bar tightens only after 50 steps, allowing the model to explore first and then exploit later. Furthermore, by filtering out low-reward trajectories, we back-prop only through "better-than-reference" samples, which reduces gradient variance without hand-tuning the gradient clip or KL penalty.

By incorporating the reference-based mechanism into Eq. (5), our final objective function is denoted as Reference-Guided GFlowNet fine-tuning (RGFN):

$$\mathcal{L}_{\text{RGFN}}(Z_i; \theta) = \sum_{i=1}^{m} \mathbb{I}(Z_i) \cdot \mathcal{L}_{\text{ISubTB}}(Z_i; \theta). \tag{7}$$

### 3.3 Bayesian Inference over Latent Rationales

Inference-scaling method such as Best-of-N (BoN) generates multiple candidate responses and select the best one based on a verifier are widely used in reasoning LVLM [47, 46]. However, BoN has significant computational overhead [18], high dependency on reward model quality [2], and scalability challenges [30]. To address these limitations, this work introduces a probabilistic method, namely **B**ayesian **i**nference over **N** latent rationales (**BiN**). Our approach is inspired by recent advancements in amortized variational inference for hierarchical models [1], where shared parameters represent local distributions, facilitating scalable inference.

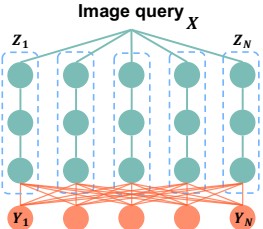

Figure 4: Inference pipeline of BiN.

Given input $X$ and a target answer $Y$, we can sample latent rationales $Z$ from a posterior $P(Z|X, Y)$ that bridges $X$ and $Y$, forming a joint sequence $XZY$. The joint likelihood is denoted as $P(XZY)$, and the marginal likelihood of $Y$ given $X$ is expressed as

$$P(Y \mid X) = \sum_{Z \sim P(Z|X,Y)} P(ZY \mid X) = \sum_{Z \sim P(Z|X,Y)} P(Z \mid X) \cdot P(Y \mid XZ). \tag{8}$$

However, it is infeasible to sample all latent rationales from the $P(Z|X, Y)$. Therefore, we employ the policy model $q_\theta(Z|X)$ trained via Eq (7) to approximate the marginal likelihood. Fig. 4 shows the complete inference pipeline where we perform the following steps: (i) Sample $N$ latent rationales $\{Z_i\}_{i=1}^{N}$ from the learned policy model: $Z_i \sim q_\theta(Z|X)$. (ii) For each sampled rationale $Z_i$, we

| $X_{system-message}$ $\langle eos \rangle$ | $X_{system-message}$ $\langle eos \rangle$ | $X_{system-message}$ $\langle eos \rangle$ |
|---|---|---|
| ***User*** : $X_{image}$ $X_{instruct}^1$ $\langle eos \rangle$ | ***User*** : $X_{image}$ $X_{instruct}^1$ $\langle eos \rangle$ | ***User*** : $X_{image}$ $X_{instruct}^1$ $\langle eos \rangle$ |
| | | ***Analyzer*** : $Z^1 \langle eos \rangle$ *(Optional)* |
| ***Assistant*** : $X_{answer}^1$ $\langle eos \rangle$ | ***Assistant*** : $\langle think \rangle$ $X_{think}^1$ $\langle /think \rangle$ \n $\langle conclusion \rangle$ $X_{answer}^1$ $\langle /conclusion \rangle$ $\langle eos \rangle$ | ***Assistant*** : $X_{answer}^1$ $\langle eos \rangle$ |
| Pre-trained LVLM | Fine-tuned reasoning LVLM | Latent reasoning LVLM (Ours) |

Figure 5: Input sequence of training a reasoning LVLM. We use token to represent learnable parts. Specifically, the fine-tuned reasoning LVLM heavily relies on annotated data during optimization, and the object tokens followed by ***Assistant*** enforce reasoning for all instructions. We introduce a new role token ***Analyzer***, so the model can selectively provide reasoning steps.

sample the corresponding answer $Y^{(i)}$ from $\pi_\Phi(Y_i|XZ_i)$, where $\pi_\Phi$ is a reasoning LVLM. (iii) Compute the joint likelihood for all pairs $(Z_iY_i)$: $\pi_\Phi(Z_iY_i|X)$. (iv) Estimate the marginal likelihood by normalizing over sequence length $|Z_iY_i|$ as

$$P(Y_i \mid X) \sim \frac{1}{N} \sum_{j=1}^{N} \frac{1}{|Z_iY_i|} \pi_\Phi(Z_iY_i \mid X). \tag{9}$$

(v) Select the answer $Y_{i*}$ with the highest estimated marginal likelihood: $i^* = \arg\max_i P(Y_i|X)$ as the final output. This inference strategy aligns with Bayesian sampling principles by approximating the marginal likelihood $P(Y|X)$ through sampling over latent rationales. The use of amortized variational inference for $q_\theta(Z|X)$ enables efficient sampling without the need for computationally intensive methods like Markov Chain Monte Carlo (MCMC). By selecting the answer with the highest estimated marginal likelihood, we aim to improve the interoperability of answer selection.

## 4 Empirical results

### 4.1 Implementation

**Reward model.** This work utilizes a fine-tuned reasoning LVLM denoted as $\pi_\Phi$ parameterized by $\Phi$ as the reward model $R$. Efficiently, $\pi_\Phi$ also acts as the starting point of the proposed rationale sampler. The purpose of our reward model is to evaluate the quality of rationales sampled from the policy model (rationale sampler). To make sure that the reward function returns a higher reward for better rationale, we first optimize $\pi_\Phi$ by maximizing the likelihood of high-quality, structured examples of rationales (SFT), such as chain-of-thought (CoT) sequences. By learning from these examples, the model gains an initial understanding of how to approach complex tasks methodically. For training $\pi_\Phi$, we consider two pre-trained LVLMs as the base models, including Qwen2.5-VL-3B & 7B [3] and a mixture of visual reasoning datasets from LLaVA-CoT [47] and R1-Onevision [48]. As shown in Fig. 5, we formulate the instructional data with a new special token ***Analyzer***. We fully fine-tune $\pi_\Phi$ using the regular token prediction loss for one epoch.

**Rationale sampler.** To sample the latent rationale $Z$ from the posterior defined in Eq.(3), we parameterize the policy model as an autoregressive model $q_\theta(Z|X)$, initialized with $\pi_\Phi$. For training, we optimize the model using *LoRA* with $r = 64$ and $alpha = 128$. We resample $3k$ visual reasoning sample from the SFT data, where each consists of (image, query, CoT, and answer). To be noted, we use the CoTs generated by teacher models, such as GPT-4o or Deepseek-R1, as our reference rationale $Z_{ref}$ in Eq. (6). For the reward approximation defined in Eq. 4, we set $\lambda = 8$ for all the experiments. Please refer to Appendix B.2 for the study of $\lambda$. More hyperparameter settings can be found in Appendix B.5.

### 4.2 Multi-modal Reasoning

**Task description.** Multi-modal reasoning evaluates the visual understanding and reasoning ability of LVLM as it requires step-by-step thinking and correct answer searching. This work proposes a reasoning LVLM, i.e., **LaCoT**, which consists of a latent rationale sampler $q_\theta$ and an answering model $\pi_\Phi$. Specifically, at test time, we randomly sample $m$ latent rationales $Z$ for an unseen $X$ with

Table 1: Test accuracy (%) on visual reasoning benchmarks. † are results based on our reproduced experiments. The best results are **bold**, and the second-best results are underlined. We choose the reasoning models fine-tuned with SFT and GRPO (R1-Onevision) as baselines. All the base models were prompted by a *step-by-step reasoning* instruction.

| Method | MathVista mini | MathVision full | MathVerse vision-only | MMMU val | MMMU-pro vision | MMVet test | MME test |
|---|---|---|---|---|---|---|---|
| GPT-4o | 60.0 | 30.4 | 40.6 | 70.7 | 51.9 | 69.1 | 2329 |
| Gemini-1.5-Pro | 63.9 | 19.2 | - | 65.8 | 46.9 | 64.0 | 2111 |
| Claude-3.5-Sonnet | 67.7 | - | 46.3 | 68.3 | 51.5 | 66.0 | 1920 |
| InternVL2-4B [7] | 58.6 | 16.5 | 32.0 | 47.9 | - | 55.7 | 2046 |
| Qwen2.5-VL-3B† [3] | 60.3 | **21.2** | 26.1 | 46.6 | 22.4 | 61.4 | 2134 |
| LaCoT-Qwen-3B | **63.2** | 20.7 | **40.0** | **48.8** | **28.9** | **69.6** | **2208** |
| LLaVA-CoT-11B [47] | 52.5 | - | 22.6 | - | - | 64.9 | - |
| LLaVA-OV-7B† [21] | 63.2 | 11.1 | 26.2 | 48.8 | 24.1 | 57.5 | 1998 |
| MiniCPM-V2.6 [49] | 60.6 | 17.5 | 25.7 | 49.8 | 27.2 | 60.0 | 2348 |
| InternVL2-8B [7] | 58.3 | 18.4 | 37.0 | 52.6 | 25.4 | 60.0 | 2210 |
| Qwen2.5-VL-7B† [3] | 63.7 | **25.4** | 38.2 | 50.0 | 34.6 | 70.5 | 2333 |
| R1-Onevision† [48] | 64.1 | 23.9 | 37.8 | 47.9 | 28.2 | 71.1 | 1111 |
| LaCoT-Qwen-7B | **68.4** | 24.9 | **43.3** | **54.9** | **35.3** | **74.2** | **2372** |

temperature $\tau$ from $q_\theta(Z|X)$, then the answer model samples $m$ answers from $\pi_\Phi(Y|XZ)$. Finally, we estimate the marginal likelihood of each answer $P(Y|X)$ using the proposed BiN and return the highest one as the final output, as shown in Eq. (9).

**Benchmarks.** This work utilizes three mathematical and one general domain reasoning benchmarks: (i) MathVista [24]: a math benchmark designed to combine challenges from diverse mathematical and visual tasks, requiring fine-grained visual understanding and compositional reasoning. (ii) MathVision [42]: a meticulously curated collection of 3,040 high-quality mathematical problems with visual contexts sourced from real math competitions. (iii) MathVerse [55]: an all-around visual math benchmark designed for an equitable and in-depth evaluation of LVLMs. We report the Vision-Only result on 788 questions, which reveals a significant challenge in rendering the entire question within the diagram. (vi) MMMU [51]: a benchmark designed to evaluate LVLM on massive multi-discipline tasks demanding college-level subject knowledge and deliberate reasoning. Furthermore, we conduct additional experiments on MMMU-pro [52], MMVet [50], and MME [9], where MMMU-Pro is a more robust version of MMMU, designed to assess LVLMs' understanding and reasoning capabilities more rigorously.

**Results.** This work provides two LaCoT models (3B and 7B). From the results summarized in Table 1, our models are the best open-source LVLM and narrow the gap to GPT-4o to less than 3 points while using only 7 billion parameters. The consistent improvements on MathVista and MMMU show that LaCoT strengthens general multi-modal reasoning. MathVerse-Vision-only improves the most, especially at 3B, where accuracy jumps 14 points and outperforms all 7B models. This advancement indicates that LaCoT significantly boosts diagram comprehension and OCR robustness. On the other hand, MathVision consists of real Olympiad diagrams, which are more varied, and often handwritten or low-resolution, conditions that push OCR and visual grounding beyond. Many problems split critical information between text and picture (e.g., tiny angle labels or subtle curve annotations), so a single misread propagates through the longer, proof-style reasoning chains, leading to a performance drop.

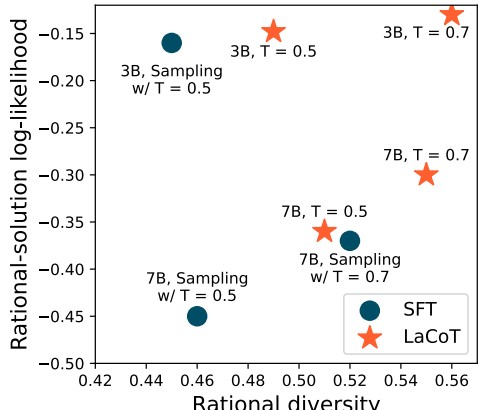

Figure 6: Maximum log-likelihood and diversity of the sampled rationale. LaCoT model (⋆) samples higher log-likelihood rationale while maintaining higher rationale diversity than SFT (●) model.

The LaCoT model can sample rationales with higher diversity than baseline models, increasing the probability of sampling answers with higher likelihood. To validate this hypothesis, we sample 5 rationale candidates with random temperature (T) for each visual instruction. To measure the semantic diversity of the samples, we compute the average inter-sentence similarity between the candidate and the reference set. As shown in Fig. 6, rationales generated by LaCoT-Qwen-3B with $T = 0.7$ have the highest log-likelihood and diversity. Qualitative results can be seen in Fig. 8 and the supplementary.

## 4.3 Inference-time Scaling

We compare BiN (ours) with Best-of-N (BoN) using LaCoT-Qwen as the shared policy model. At inference, we sample $N$ rationale–answer pairs, compute a length-normalized log-likelihood of each answer as the reward, and for BoN select the answer with the highest reward. To ensure fairness, no external reward model is used. We evaluate $N \in \{5, 10\}$ for both methods and report the best score per method. As shown in Table 2, BiN consistently outperforms BoN on visual reasoning benchmarks.

Table 2: Comparison between two inference-time scaling methods using LaCoT-Qwen (3B/7B).

| Method | MathVerse | MathVista | MMMU | MMVet |
|---|---|---|---|---|
| 3B w/ BoN | 21.2 | 57.1 | 44.7 | 67.1 |
| 3B w/ BiN (ours) | **40.0** | **63.2** | **48.8** | **69.6** |
| 7B w/ BoN | 26.5 | 62.2 | 47.3 | 71.2 |
| 7B w/ BiN (ours) | **39.7** | **68.4** | **54.9** | **74.2** |

## 4.4 Ablation Studies

**Effectiveness of RGFN.** As baselines, we consider zero-shot prompting w/o reasoning, supervised fine-tuning on the visual reasoning dataset, and GRPO [35] fine-tuning.

From the results summarized in Table 3, the base model performs well without chain-of-thought reasoning. While supervised fine-tuning on reasoning data slightly improves performance on two benchmarks, it still struggles to generalize to challenging visual reasoning tasks. Fine-tuning with GRPO yields poor performance, partly due to inadequate guidance of the external reward model, i.e., it cannot distinguish good rationales from bad ones, and limited exploration due to the KL penalty. Such misleading

Table 3: Test accuracy (%) on reasoning benchmarks using Qwen2.5-VL-7B model.

| Method | MathVista | MathVerse | MMMU |
|---|---|---|---|
| Zero-shot | 63.7 | 38.2 | 50.0 |
| SFT | 62.7 | 38.7 | 50.6 |
| GRPO | 62.6 | 36.8 | 47.9 |
| RGFN | **68.4** | **43.3** | **54.9** |

optimization due to misaligned reward is a widely noted issue in RL-based algorithms [57] for LVLM. On the other hand, by matching the target distribution, RGFN avoids collapsing to a single mode of the reward, and the reference-guided exploration covers diverse trajectories, leading to better performance on complex examples.

**Study of BiN.** To evaluate the scalability, we apply the proposed inference-scaling method by varying the number of candidates $N$ and temperature $T$ using LaCoT-Qwen-3B on the reasoning benchmarks. As illustrated in Fig. 7, test accuracy consistently increases with higher $N$, and higher $T$. This indicates that increasing $N$ systematically improves test-time accuracy because it (i) reduces the Monte-Carlo variance of the marginal-likelihood estimator—standard error scales as $\mathcal{O}(1/\sqrt{N})$—thereby stabilizing answer rankings; (ii) offers broader pos-

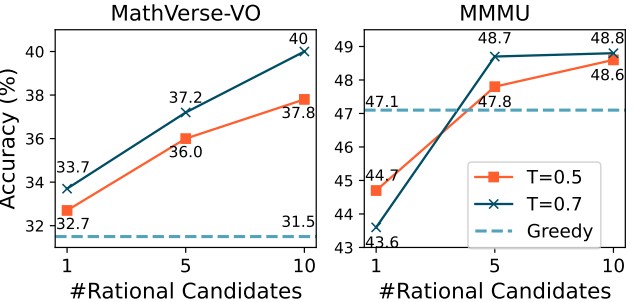

Figure 7: Test accuracy on reasoning benchmarks using LaCoT-Qwen-3B. We evaluate the impact of #rationale candidates ($N$) and random temperature (T).

terior coverage, mitigating mode-dropping bias inherent in the amortized sampling $q_\theta(Z|X)$; (iii) smooths fluctuations introduced by length-normalization, yielding more reliable re-weighting; and (iv) enlarges the candidate answer set, elevating the chance that the correct output is observed. Together, these effects drive an exponential decay in the probability of selecting an incorrect answer.

Furthermore, higher $N$ can effectively address the hallucination issue in visual reasoning. As shown in Fig. 7, when the sampled rational size $N = 1$, BiN may produce incorrect or misleading reasoning steps and lead to lower answer accuracy on the MMMU dataset. However, increasing $N$ from 1 to 5 significantly mitigates hallucination and improves answer accuracy. We provide qualitative results in Appendix B.1.

To evaluate the generalizability of BiN, we evaluate the performance of Qwen2.5-VL 3B & 7B (SFT) with the proposed inference-scaling method. We set $N = 5$ and $T = 0.7$, which gives the most performance boost with a relatively shorter inference time. As shown in Table 4, BiN consistently improves the model performance on all benchmarks, indicating the effectiveness of BiN as a general inference-scaling method for reasoning LVLMs.

Table 4: Test accuracy of Qwen2.5-VL supervised fine-tuning on reasoning data.

| Method | MathVista | MathVerse | MMMU |
|---|---|---|---|
| 7B (SFT) | 62.7 | 38.7 | 50.6 |
| + BiN | **64.4** | **38.9** | **51.6** |
| 3B (SFT) | 58.7 | 33.3 | 43.1 |
| + BiN | **59.4** | **35.2** | **45.0** |

## 5   Related Work

**Learning-based Multimodal CoT (MCoT)** methods have emerged as a powerful paradigm for enhancing the reasoning capabilities of LVLMs [40, 39]. Unlike prompt-based or plan-based approaches, learning-based MCoT explicitly embeds the entire reasoning trajectory into the models through supervised learning on rationale-augmented datasets. Early studies such as Multimodal-CoT [56] pioneered this direction by fine-tuning LVLMs to generate visual CoT, facilitating a structured reasoning process aligned with human cognitive patterns. From that, methods like MC-CoT [46] further refined this approach by incorporating multimodal consistency constraints and majority voting mechanisms during training. In addition, methods such as PCoT [43] and G-CoT [25] demonstrated that explicitly training LVLMs with structured rationales improves the interpretability and generalizability. These advancements underscore the effectiveness and necessity of embedding structured, rationale-driven reasoning capabilities directly into multimodal models.

**Reinforcement Learning-based Language Models** have demonstrated significant effectiveness in advancing the reasoning capabilities of LLMs. DeepSeek-R1 [12] exemplifies this by activating long-chain-of-thought (long-CoT) reasoning solely through reinforcement learning (RL), achieving improvements over models such as GPT-o1 [17] in specific aspects when combined with supervised fine-tuning (SFT) cold starts and iterative self-improvement. This success has spurred further interest in RL-driven models, including Open-R1 [8] and TinyZero [31]. To enhance reasoning, generalization, and ensure training stability, several RL algorithms have been developed, such as PPO [34], GRPO [12], and simplified methods like RLHF [58], DPO [32], and SPO [38]. Nevertheless, these approaches are heavily dependent on high-quality human-annotated data (e.g., human preference labels and scalar rewards) and typically produce policies with limited diversity. To address these limitations, this work proposes an RL algorithm specifically designed to train LVLMs using amortized variational inference, which is capable of generating diverse outputs and supporting probabilistic inference-time scaling.

**Inference-time Scaling methods** aim to enhance reasoning performance during inference by leveraging high-quality prompts and effective sampling strategies. Plan-based approaches, exemplified by MM-ToT [10] and LLaVA-CoT [47], utilize search strategies such as DFS and BFS, including Best-of-N search, sentence-level beam search, and stage-level beam search, to identify optimal reasoning trajectories. These methods typically assess candidate trajectories using scalar metrics ranging from 0.1 to 1.0. However, such explicit evaluation is computationally expensive, as each candidate requires an additional forward pass through a dedicated reward model. To mitigate this computational overhead, our work introduces a learning-based algorithm designed to align the marginal likelihood of generating a rationale directly with its reward. This approach enables efficient probabilistic sampling without explicit reward computations during inference.

## 6   Conclusion

In a real-world scenario, solving a fixed visual query with different reasoning chains that lead to the correct answer requires a nuanced understanding of image, context, logic, and flexibility in

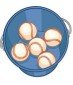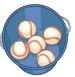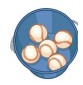

**Question: How many baseballs are there?**

**Reasoning**: The image shows four blue buckets, each containing five baseballs. Therefore, the total number of baseballs is calculated by multiplying the number of buckets by the number of baseballs per bucket.
***4 buckets × 5 baseballs/bucket = 20 baseballs.***
**Final output**: **There are 20 baseballs.**

**GRPO**

**Reasoning**: The image shows four blue buckets, each containing five baseballs. To determine the total number of baseballs, we can directly count the baseballs in each bucket. As seen in the image, each bucket has five baseballs. Since there are four buckets and each contains five baseballs, we can calculate the total by multiplying the number of buckets by the number of baseballs per bucket:
***4 buckets × 5 baseballs per bucket = 20 baseballs.***
There is no need to consider any additional factors or hidden details since the image clearly shows the contents of each bucket. *Higher diversity*
**Final output**: **20**

**LaCoT**

Figure 8: Qualitative results of visual reasoning. LaCoT can sample a more diverse and comprehensive reasoning chain than the GRPO model.

thought. While querying this knowledge in LVLM involves sampling from intractable posterior distributions. To address this challenge, we propose a novel training algorithm based on amortized variational inference for latent visual chains-of-thought (CoT). Our approach incorporates **token-level reward approximation** and **RGFN**, enabling effective and efficient optimization of a policy model to generate diverse and plausible reasoning trajectories, outperforming both supervised fine-tuning and reward-maximization baselines. In addition, we introduce a new inference-time scaling strategy, **BiN**, which mitigates reward hacking and enhances interpretability with statistically robust selection criteria. Building upon these components, we present **LaCoT** that leverages a rationale sampler for general-purpose visual reasoning, and an answer generator that is enhanced by high-quality reasoning chains. Given this system, future work should investigate the possibility of applying it for knowledge distillation and synthetic data generation.

**Limitations.** Due to resource constraints, we apply the proposed methods to models up to 7B parameters, but we expect the conclusions to hold for larger models. In fact, our training and inference method can be applied to any autoregressive model, including LLM and LVLM, with various model sizes. As with any on-policy method, exploration in tasks with complex latent remains an open challenge since multiple factors can affect the exploration time, such as sequence length and technical challenges like memory cost. Despite the improved inference performance, this work does not address issues such as hallucination, which are closely related to internal knowledge.

## Acknowledgments and Disclosure of Funding

This research was supported in part by the DEVCOM Army Research Laboratory under Contract W911QX-21-D-0001, the National Science Foundation under Grant 2502050, and the National Institutes of Health under Award R16GM159146. The content is solely the responsibility of the authors and does not necessarily represent the official views of the funding agencies.

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

## A   Proof of Proposition 1

*Proof.* **(a)** Assume that within the segment $\{t, t+1, \ldots, t+\lambda\}$ the true reward grows linearly, i.e.

$$R(z_{1:t+i}\top) \;=\; R(z_{1:t}\top) \;+\; i\,\Delta, \qquad \Delta := \frac{R(z_{1:t+\lambda}\top) - R(z_{1:t}\top)}{\lambda}, \quad 0 \le i \le \lambda.$$

Substituting this expression into Eq. (4) shows $\tilde{R}(z_{1:t+i}\top) = R(z_{1:t+i}\top)$ for every $i$, so the interpolation incurs *zero* error.

**(b)** Suppose $R$ is twice–differentiable along the trajectory and its discrete second derivative is bounded:
$$\big| R(z_{1:s+1}\top) - 2R(z_{1:s}\top) + R(z_{1:s-1}\top) \big| \;\le\; M, \qquad \forall s.$$
The classical linear–interpolation error bound then yields

$$\big| \tilde{R}(z_{1:t+i}\top) - R(z_{1:t+i}\top) \big| \;\le\; \frac{M}{8}\, i(\lambda - i) \;\le\; \frac{M\lambda^2}{8}, \qquad 0 \le i \le \lambda. \tag{10}$$

Thus the approximation error decays as $\mathcal{O}(\lambda^2)$; choosing $\lambda$ sufficiently small keeps it arbitrarily close to 0.

Let

$$F(z_s) \;:=\; \frac{R(z_{1:s}\top)}{q_\theta(\top \mid z_{1:s})}, \qquad \tilde{F}(z_s) \;:=\; \frac{\tilde{R}(z_{1:s}\top)}{q_\theta(\top \mid z_{1:s})}, \qquad \varepsilon_s := \tilde{R}(z_{1:s}\top) - R(z_{1:s}\top).$$

From Eq. (10) we have $|\varepsilon_s| \le \frac{M\lambda^2}{8}$, so

$$\tilde{F}(z_s) = F(z_s)\Big(1 + \frac{\varepsilon_s}{R(z_{1:s}\top)}\Big).$$

Denote $R_{\min} := \min_{s \le t+\lambda} R(z_{1:s}\top) > 0$ (positivity follows from likelihoods). Then $\big|\varepsilon_s/R(z_{1:s}\top)\big| \le \frac{M\lambda^2}{8R_{\min}}$.

Consider any $i < j \le t + \lambda$. Applying Eq. (1) to both $F$ and $\tilde{F}$, we obtain

$$\tilde{F}(z_i) \prod_{k=i+1}^{j} P_F(z_k \mid z_{k-1}) = \tilde{F}(z_j) \prod_{k=i+1}^{j} P_B(z_{k-1} \mid z_k)\Big(1 + \mathcal{O}(\lambda^2)\Big),$$

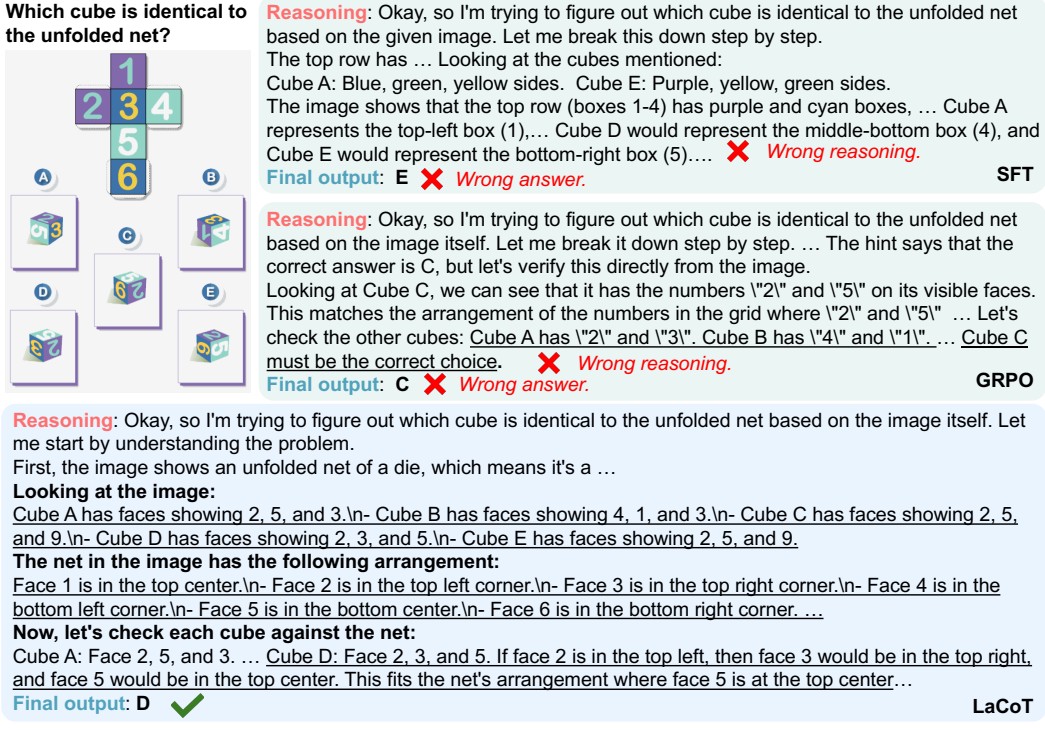

Figure F9: Qualitative results of visual reasoning. We **highlight** the important reasoning steps.

where the $\mathcal{O}(\lambda^2)$ term accumulates at most $(j - i)$ relative perturbations bounded by $M\lambda^2/(8R_{\min})$. Hence, the forward and backward product flows still match up to a multiplicative factor that vanishes quadratically as $\lambda \to 0$. Therefore, the interpolated rewards preserve flow consistency to arbitrary precision for sufficiently small segment lengths. $\square$

# B    Experiments

## B.1    Qualitative results

In Fig. F9, we provide qualitative results of a comparison between Qwen2.5-VL-7B (SFT), Qwen2.5-VL-7B (GRPO), and LaCoT-Qwen-7B. As can be seen, LaCoT-Qwen-7B can provide a more accurate reasoning chain, leading to the correct answer. Meanwhile, due to limited generalizability, SFT and GRPO samples show the wrong visual CoT. In Fig. F10 and Fig. F11, our LaCoT model can sample more straightforward and accurate reasoning chains, demonstrating the effectiveness and robustness of the proposed training and inference algorithm.

## B.2    Study of interpolation reward

In Table T5, we study the impact of the interpolation reward with different skipped steps (i.e., $\lambda$) in the reward approximation process of the policy optimization. As mentioned in Proposition 1, a smaller $\lambda$ theoretically leads to more fine-grained reward supervision but longer training time.

## B.3    Efficiency analysis

Give superior performance gain by sampling multiple rationales at inference time, but this process introduces additional inference cost, and we address this by using mini-batching (with batch size k=5) to generate N rationales in N/k forward passes. In Table T6, we report the average per-sample inference time (reasoning + answering) and corresponding performance of different reasoning-LVLM on MathVista and MathVerse. As can be seen, LaCoT-Qwen-7B consistently achieves stronger

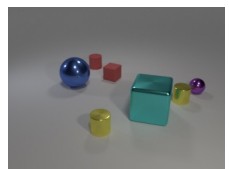

**Question: Subtract all green rubber cubes. Subtract all red matte blocks. How many objects are left?**

**Reasoning**: Okay, so I need to figure out how many objects are left after subtracting all green rubber cubes and red matte blocks from this image. Let me start by understanding what each term means.
First, "green rubber cubes." The image shows three cubes: one blue with a reflective surface, one cyan with a glossy finish, and one yellow with a shiny appearance. **There is no green cube in the image. Therefore, no green rubber cubes exist to remove.**
Next, "red matte blocks." The image shows two small red objects, both of which appear to be cubes based on their shape. They do not look matte; they have a glossy finish. Therefore, there are **no red matte blocks to subtract either**. ❌ *Wrong reasoning.*
Given that neither green rubber cubes nor red matte blocks are present in the image, we can conclude that **no objects need to be removed from the total count.**
Looking at the initial setup described: there\'s a large blue sphere with a reflective surface, a smaller red cylinder, a medium-sized red matte cube, a blue-tinted metallic cube, a small yellow cylinder, another small yellow cylinder, and a small purple sphere. Let\'s count them:
1. Blue sphere (reflective)\n2. Red cylinder (cylinder)\n3. Red matte cube (cube)\n4. Blue-tinted metallic cube (cube)\n5. Yellow cylinder (cylinder)\n6. Another yellow cylinder (cylinder)\n7. Purple sphere (sphere)
That makes seven objects in total. Since neither green rubber cubes nor red matte blocks were present in the image, all seven objects remain.
Therefore, the answer should be 7.
**Final output**: **7.** ❌ *Wrong answer.* **SFT**

**Reasoning**: The image contains several objects: a blue sphere, two red cylinders, a red cube, a teal cube, a yellow cylinder, a purple sphere, and another yellow cylinder.
**First, I need to identify and remove any green rubber cubes. As seen in the image, there are no green rubber cubes present.**
**Next, I will look for any red matte blocks. The image shows a red cube, which appears to be matte. I will subtract this red cube from the total count.** ✅
Now, let's count the remaining objects:\n- Blue sphere\n- Two red cylinders\n- Red cube (removed)\n- Teal cube\n- Yellow cylinder\n- Purple sphere\n- Another yellow cylinder
After removing the red cube, the remaining objects are:\n1. Blue sphere\n2. Two red cylinders\n3. Teal cube\n4. Yellow cylinder\n5. Purple sphere\n6. Another yellow cylinder
This totals to six objects.
Therefore, after removing the red block, six objects remain ✅
**Final output**: **6** ✅ **LaCoT**

Figure F10: Qualitative results of visual reasoning. We **highlight** the important reasoning steps.

Table T5: Study the impact of the interpolation reward with different skipping steps (i.e., $\lambda$) to the policy model.

| Method | $\lambda$ | MathVista mini | MathVision full | MathVerse vision-only | MMMU val | Overall *Avg.* |
|---|---|---|---|---|---|---|
| Qwen2.5-VL-7B | - | 63.7 | **25.4** | 38.2 | 50.0 | 44.3 |
| LaCoT-Qwen-7B | 32 | 64.9 | 23.0 | 42.5 | 51.9 | 45.6 |
| LaCoT-Qwen-7B | 8 | **68.4** | 24.9 | **39.7** | **54.9** | **47.0** |

performance, even with modest increases in inference time. Compared to other multi-rationale baselines, LaCoT strikes a favorable balance between computational cost and reasoning reliability, thereby improving both the trustworthiness of rationales and the accuracy of final answers.

## B.4 Experiments compute resources

This work utilizes an 8*80GB GPU-node for training. We set the Deepspeed Zero-3 stage and gradient-checkpointing to reduce memory costs during optimization. It takes around 30 hours for supervised fine-tuning on 250k reasoning data samples, and 120 hours for GRPO and RGFN fine-tuning on 3k data samples.

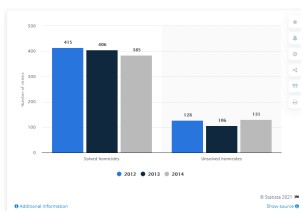

**Question: What is the average of longest light blue bar and shortest gray bar?**

**Reasoning**: Okay, so I need to figure out the average of the longest light blue bar and the shortest gray bar from the given image. Let me start by understanding what each part means.
First, the image shows a bar chart comparing solved and unsolved homicides over three years: 2012, 2013, and 2014. The bars are color-coded—light blue for 2012, dark blue for 2013, and gray for 2014.
Looking at the data provided in the Python code, there\'s a list called `homicide_data` which includes two arrays: `solved` and `unsolved`. The `solved` array has values [415, 406, 385] corresponding to the years 2012, 2013, and 2014 respectively. As seen in the image, the light blue bars represent 2012, the dark blue bars represent 2013, and the gray bars represent 2014. So, the light blue bar is for 2012 with a value of 415.
Next, the `unsolved` array has values [128, 106, 131]. The first value (128) is for 2012, the second (106) is for 2013, and the third (131) is for 2014. However, since the question mentions solving or unsolving homicides, it seems we\'re focusing on the solved ones here because the user referred to "longest light blue bar." But wait, **the light blue bar is actually the solved homicides for 2012, which is 415.** ✅
Now, the gray bar is for 2014, which is the shortest among all the gray bars. As seen in the image, the gray bars represent 2014, and the shortest one is 385, but the note says it\'s 385? Wait, no—the note mentions that 385 is for 2014, but looking back, the `unsolved` array has 385 as its last value, which corresponds to 2014. So perhaps the gray bar is indeed the shortest unsolved homicides, which would be 385. But let me check again. Wait, maybe I\'m mixing up something. Let me clarify. The solved homicides are shown in light blue, and their values are [415, 406, 385]. The unsolved homicides are shown in gray, with values [128, 106, 131]. So for 2014, both solved and unsolved have numbers, but the gray bar specifically refers to unsolved homicides, which is lower than the solved ones. So the shortest gray bar is 106 (for 2013).
But hold on, the user mentioned the longest light blue bar and the shortest gray bar. So the light blue bar is 415, which is the longest. The gray bar varies between 131, 385, and 106. The smallest number here is 106, which is in 2013. So **the shortest gray bar is 106.** ❌ *Wrong reasoning.*
So now, averaging these two: 415 and 106. To find the average, I add them together and divide by 2. That gives **(415 + 106)/2 = 521/2 = 260.5**. Since the user asked for an integer, I should round this. 260.5 rounds to 261. So the average is 261.
**Final output**: **261.** ❌ *Wrong answer.*                                           **SFT**

**Reasoning**: Okay, so I need to figure out the average of the longest light blue bar and the shortest gray bar in the given image.
First, let\'s look at the image itself. The image shows a bar chart with two main categories: "Solved homicides" and "Unsolved homicides." Each category has three bars representing the years 2012, 2013, and 2014. The colors used are light blue for 2012, dark blue for 2013, and gray for 2014.
Now, focusing on the light blue bars, which represent the year 2012. The values for solved homicides are 415, and for unsolved homicides, it\'s 128. So, the light blue bars show 415 for solved and 128 for unsolved.
Next, looking at the gray bars, which represent the year 2014. The values here are 385 for solved homicides and 131 for unsolved. So, the gray bars show 385 and 131.
The question asks for the average of the longest light blue bar and the shortest gray bar. **The longest light blue bar would be the one with the highest value, which is 415 for solved homicides. The shortest gray bar would be the one with the lowest value, which is 131 for unsolved homicides.** ✅
So, we have the numbers 415 (longest light blue) and 131 (shortest gray). To find the average, we add these two together and divide by 2.
Calculating that: **\\(415 + 131 = 546\\), then \\(546 / 2 = 273\\).** ✅
Therefore, the average should be 273.
**Final output**: **273** ✅                                                          **LaCoT**

Figure F11: Qualitative results of visual reasoning. We **highlight** the important reasoning steps.

## B.5  Hyperparameter

We detail the hyperparameters used for training the reward model and LaCoT in our experiments in Table T7. During LaCoT training, we randomly sample (mini-batch size) $Z$s for every $(X, Y)$ as exploration.

Table T6: Inference time study of reasoning model with multiple rational sampling.

| #Rationals (N) | 1 | 5 | 10 | MathVista | MathVerse |
|---|---|---|---|---|---|
| LLaVA-CoT-11B | - | 340s | 830s | 52.5 | 22.6 |
| R1-OneVision-7B | 32s | - | - | 64.1 | 37.8 |
| LaCoT-Qwen-7B | - | 30s | 65s | 68.4 | 39.7 |

Table T7: Hyperparameters for training.

| | |
|---|---|
| LoRA dropout | 0.05 |
| Batch size (SFT) | 2 |
| Batch size (RGFN) | 1 |
| Gradient accumulation (SFT) | 16 |
| Learning rate | 0.00001 |
| Optimizer | AdamW |
| Weight decay | 0.05 |
| Temperature max | 1.0 |
| Temperature min | 0.5 |
| Reward temperature start | 1.0 |
| Reward temperature end | 0.7 |
| Reward temperature horizon | 50 |
| exploration number | 6 |
| $\lambda$ | 8 |
| $\tau_{max}$ | 1.5 |
| $\tau_{min}$ | 1.0 |
| Maximum rationale length | 700 |
| Minimum rationale length | 64 |

