# OpenReview forum: "Latent Chain-of-Thought for Visual Reasoning"
_NeurIPS.cc/2025/Conference — NeurIPS 2025 poster_

### Official Review · Reviewer_AnFV · 2025-06-27

**Clarity:** 3
**Significance:** 3
**Originality:** 3
**Rating:** 5
**Confidence:** 4

**Summary:**

The authors propose an RL algorithm to improve on existing CoT reasoning methods for LVLMs as current popular methods, SFT & RL approaches such as PPO and GRPO, fail to generalise well and may rely on noisy or biased reward models.  SFT models tend to struggle with tasks out-of-distribution while the RL approaches are constrained in exploration and often fail to discover novel and useful rationales.  They use theory from Generative Flow Networks (section 2 & 3) to do amortized variational inference (AVI), in effect casting visual reasoning as a variational inference problem, to generate sequences of latent rationales while also introducing a token-level reward approximation that they guide their training approach with, a method they call Reference-Guided GFlowNet fine-tuning (RGFN - section 3.2).

The overall aim is improved generalisation on out-of-distribution vision-languague tasks by way of RGFN to induce exploration of better rationales.  At inference time they combine this with a  bayesian inference method (BiN) for sampling latent rationales in contrast to Best-of-N and Beam search methods which incur a high computational burden, are difficult to scale, and rely on reward models (section 3.3). The resulting model is the LaCoT trained from open pre-trained models (Qwen2.5-VL-3B & 7B) on a mixture of visual reasoning datasets.

Key to this approach (section 3.1) is token level reward modeled on the joint likelihood of the latent rationales with the question/answer pairs (X,Y) where linear interpolation provides reward estimates for latent tokens along the sequence. During training a reference rationale helps guide training toward better rationales vis-a-vis the rationale sampler and reward model (section 3.2).

In section 4 the approach is evaluated where LoRa is used to finetune the Qwen2.5-VL open models.  In 4.2 LaCot is compared to SFT & GRPO baselines on Math and reasoning benchmarks: MathVista, MathVerse, MMMU. LaCot archives SOTA (Table 1) among the open source models in these tasks, boosting diagram comprehension and OCR robustness while also yielding better diversity and generalization through its proposed rationales.  Further ablations demonstrate that SFT and GRPO fail to generalise well to more complex tasks and especially in cases where the reward model is bad or the KL limits effective exploration.

**Questions:**

I couldn't see where the reference rationales come from?  Could you explain that?  If it hasn't been included would it be worth briefly touching on this?

Did you have any issues with hallucinations in your rationales during inference?  Presumably these would be more problematic as the sample size is lowered in BiN?

When doing token level reward approximation I was a bit uncertain where rewards we actually be sampled vs. interpolated from Eq. 4, can you expand on this a bit?

**Ethical Concerns:**

["NO or VERY MINOR ethics concerns only"]

**Final Justification:**

This work addresses generalisation over problems that require reasoning in LVLMs which is a relevant and challenging issue with a focus on highly referenced mathematics benchmarks. This paper presents the problem in terms of variational inference and combines a range of techniques into a novel methodology which the authors effectively apply across empirical benchmarks.  The work itself is compelling and could feasible lead to extensions that further explore generative rationales in multimodal foundation models.

**Limitations:**

Yes.

**Paper Formatting Concerns:**

None.

**Quality:**

3

**Strengths And Weaknesses:**

**Strengths**

Generating useful rationales from visual queries with LVLMs is an open area of research and currently is the focus of ongoing efforts to drive improvements to multimodal foundation models [1,2].  Rationalising over both vision and language modalities adds complexity to the problem of generating admissible CoT while avoiding hallucinations by the model  which will need to be overcome in the methods proposed in this paper.  Reasoning well from visual observation will shed light on how to approach problems in multimodal decision making, visual scene understanding, and inferring insights from abstract imagery, all of which require strong abstraction and generalisation priors to be successful.

This work leverages a set novel strategies with that aim to generate useful rationales on which a generative model can be conditioned such that scores on reasoning benchmarks are improved overall. The authors focus on good generalisation, constraining model exploration to avoid catastrophic forgetting though fine tuning, and an inference scaling method that draws on the variational inference model.  These yield the following improvements over prior approaches:

 * The use and adaptation of Generative Flow Networks to cast the problem of generating rationales in terms of amortized variational inference.
 * Token level reward approximation based on the joint likelihood of the proposed rationales.   Under the assumption that the reward is locally smooth the authors devise a method to save costly computations via linear interpolation of rewards corresponding to latent states.  * Overall this works to improve sampling diversity of the rationales and helping to explore during training.
 * Bayesian Inference to draw latent samples from the target posterior at inference time which in favour Best-of-N and Beam Search methods which are computationally costly, are difficult to scale, and rely on reward models.  This approach extends natural from the AVI framework used in LaCoT.
 * Policy exploration is constrained by reference latent to overcome catastrophic forgetting and encourage more diversity in rationale exploration.

Overall flow of the paper is good, although I do have  few critiques on this also here which I will address below.  The big positives for me are the thorough explanation of the theory behind the overall approach and the creative way that the authors have brought together a range of methods to tackle a difficult problem.  The results presented in tables 1 & 2 and figures 5 & 6 demonstrate the efficacy of this approach considering the inference method, the number of rationales sampled, ablating RFGN, and overall reasoning scores over a wide range of models.

**Weaknesses**

As mentioned above, while I found some of the aspects of this paper clearly presented and compelling, I think some parts could be made clearer.

The biggest challenge that I had when reading this paper was that, although there are a number of good ideas put into the approach, it took me a few passes to really see how each component tied together into the holistic approach. I think this could possibly be addressed through some high level treatment of your approach, perhaps detailed at the beginning of section 3 with the potential inclusion of a new figure and that underlines the key fundamental strengths of the approach that you propose.

Then in Section 3.3 I see the approach in general however I think it could benefit from a bit more detail in the figures and main text.  Also some expansion or more description of figure 3 could help, perhaps  complemented by an algorithmic element and some explanation (if I haven't missed it) as to whether we can expect higher quality reationales from BiN when comparing to other inference methods - maybe somecomparisons of concrete examples.

One nitpick, In the intro on the second to last paragraph you mention three points that overlap with your main contributions.  Could you simply include this all under one list? It may make things a little clearer.

Finally, and this is probably out of scope for this work, but it may strengthen this work to consider LaCoT on benchmark domains targeting decision making and acting responses (e.g. E3VQA, MMIU [3,4]) that rely on some scene understanding and reasoning from world priors.

[1] Yukun Qi, Yiming Zhao, Yu Zeng, Xikun Bao, Wenxuan Huang, Lin Chen, Zehui Chen, Jie Zhao, Zhongang Qi, and Feng Zhao. 2025. VCR-Bench: A Comprehensive Evaluation Framework for Video Chain-of-Thought Reasoning. arXiv preprint arXiv:2504.07956 (2025).

[2] Guanghao Zhou, Panjia Qiu, Cen Chen, Jie Wang, Zheming Yang, Jian Xu, and Minghui Qiu. Reinforced mllm: A survey on rl-based reasoning in multimodal large language models. arXiv preprint arXiv:2504.21277, 2025.

[3] Fanqing Meng, Jin Wang, Chuanhao Li, Quanfeng Lu, Hao Tian, Jiaqi Liao, Xizhou Zhu, Jifeng Dai, Yu Qiao, Ping Luo, et al. Mmiu: Multimodal multi-image understanding for evaluating large vision-language models. arXiv preprint arXiv:2408.02718, 2024

[4] I Lee, W Park, J Jang, M Noh, K Shim, B Shim. Towards Comprehensive Scene Understanding: Integrating First and Third-Person Views for LVLMs. arXiv preprint arXiv:2505.21955, 2025•

---

> ### Author Rebuttal · Authors · 2025-07-31
>
> We thank the reviewer for their thoughtful comments and constructive suggestions. Below, we provide detailed responses to each point.
> > **`W4-1`**: Perhaps detailed at the beginning of section 3 with the potential inclusion of a new figure and that underlines the key fundamental strengths of the approach that you propose...; Then in Section 3.3 I see the approach in general however I think it could benefit from a bit more detail in the figures and main text. ...; One nitpick, In the intro on the second to last paragraph you mention three points that overlap with your main contributions. Could you simply include this all under one list? It may make things a little clearer.
>
> - We will add a figure at the beginning of section 3 to clearly show our motivation.
> - Figure 4 visualizes Equation 8, illustrating how BiN performs Bayesian marginalization over latent chain-of-thought trajectories Z to approximate the likelihood of the answer P(Y|X). We agree that additional detail would improve clarity, and we will revise both the figure and the accompanying explanation in the final version to better highlight the role of latent reasoning and its connection to our overall inference pipeline.
> - We will revise the introduction and consolidate the overlapping points into a unified contribution list to improve clarity and readability in the final version.
>
> > **`W4-2`**: Consider LaCoT on benchmark domains targeting decision making and acting responses (e.g., E3VQA[1], MMIU[2]) that rely on some scene understanding and reasoning from world priors.
>
> We appreciate the suggestion. We attempt to evaluate on **MMIU**, but find that LaCoT and our baseline models (e.g., Qwen2.5-VL) struggle to generalize due to their limited training on single-image reasoning tasks. Regarding **E3VQA**, the dataset was not publicly available at the time of writing, which prevented us from conducting a fair comparison.
>
> Although we are unable to follow the reviewer's suggestion, we conducted additional experiments on three other visual understanding benchmarks: **MMMU-pro**, **MMVet**, and **MME**. These benchmarks cover a broad range of tasks, including visual commonsense, fine-grained recognition, and multi-choice QA.
>
> | Method                | MMMU-pro | MMVet | MME  |
> |----------------------|----------|-------|------|
> | InternVL2-4B         | -        | 55.7  | 2046 |
> | Qwen2.5-VL-3B        | 22.4     | 61.4  | 2134 |
> | **LaCoT-Qwen-3B**    | **28.9** | **69.6** | **2208** |
> | InternVL2-8B         | 25.4     | 60.0  | 2210 |
> | Qwen2.5-VL-7B        | 34.6     | 70.5  | 2333 |
> | R1-Onevision         | 28.2     | 71.1  | 1111 |
> | **LaCoT-Qwen-7B**    | **35.3** | **74.2** | **2372** |
>
> These results demonstrate that LaCoT consistently improves performance across different domains and model scales, suggesting strong generalization beyond mathematical reasoning tasks.
>
>
> > **`Q4-1`**: I couldn't see where the reference rationales come from? Could you explain that? If it hasn't been included would it be worth briefly touching on this?
>
> Each training sample in our dataset consists of a tuple {image, query, CoT, answer}, where the CoT (Chain-of-Thought) serves as the reference rationale for the answer. These rationales are generated by teacher models such as **GPT-4o** or **Deepseek-R1**, depending on the data source. We will clarify this detail in the revised version of the paper for better transparency.
>
> > **`Q4-2`**: Did you have any issues with hallucinations in your rationales during inference? Presumably these would be more problematic as the sample size is lowered in BiN?
>
> We appreciate the reviewer’s valuable comment. We observed some hallucination issues in generated rationales when the sample size N=1, where BiN can occasionally produce incorrect or misleading reasoning steps on the **MMMU** dataset.
>
> To assess this, we used **one-shot reasoning** as a baseline and varied the number of sampled rationales N during inference. As shown below, increasing N from 1 to 5 significantly mitigates hallucination and improves answer accuracy:
>
> | MathVerse     | 1    | 5    | 10   |
> |---------------|------|------|------|
> | one-shot reasoning     | 31.5 | -    | -    |
> | LaCoT-Qwen-3B | 33.7 | 37.2 | 40.0 |
>
> | MMMU          | 1    | 5    | 10   |
> |---------------|------|------|------|
> | one-shot reasoning     | 47.1 | -    | -    |
> | LaCoT-Qwen-3B | 44.7 | 48.7 | 48.8 |
>
> These results suggest that **hallucination can indeed be more pronounced at smaller N**, but BiN’s sampling strategy is effective in addressing it as N increases. We will include additional qualitative examples in the final version to illustrate this effect.
>
> > **`Q4-3`**: When doing token level reward approximation I was a bit uncertain where rewards we actually be sampled vs. interpolated from Eq. 4, can you expand on this a bit?
>
> We compute the actual reward for every $\lambda$ steps, and apply linear interpolation to approximate the intermediate token-level rewards.
> This design reflects a trade-off between accuracy and efficiency: while $\lambda$=1 (i.e., no interpolation) would yield slightly better performance, it is computationally prohibitive in practice. Under our current setting, computing token-wise rewards with $\lambda$=1 takes **~50 minutes per sample due to the long visual CoT sequences**. This requires **~400 hours of training for one epoch** on a 3k-sample dataset using 8*80G GPUs. Thus, our interpolated approximation provides a practical and effective alternative that maintains strong performance while significantly reducing training time.
>
>
>
>
> [1] Fanqing Meng, Jin Wang, Chuanhao Li, Quanfeng Lu, Hao Tian, Jiaqi Liao, Xizhou Zhu, Jifeng Dai, Yu Qiao, Ping Luo, et al. Mmiu: Multimodal multi-image understanding for evaluating large vision-language models. arXiv preprint arXiv:2408.02718, 2024
>
> [2] I Lee, W Park, J Jang, M Noh, K Shim, B Shim. Towards Comprehensive Scene Understanding: Integrating First and Third-Person Views for LVLMs. arXiv preprint arXiv:2505.21955, 2025

---

> > ### Comment · Reviewer_AnFV · 2025-08-04
> > **Response to Rebuttal**
> >
> > Thanks very much for your detailed rebuttal and clarifications.  I'll maintain my score, very nice work!

---

> > > ### Author Response · Authors · 2025-08-05
> > >
> > > We are sincerely grateful for your continued support for the acceptance of this paper. Following your suggestions, we will incorporate our responses to your points into the revised version of the paper.
> > >
> > > Thank you again for your time and support!

---

### Official Review · Reviewer_tynq · 2025-06-30

**Clarity:** 3
**Significance:** 3
**Originality:** 3
**Rating:** 5
**Confidence:** 3

**Summary:**

Thia paper propose a novel  approach for COT reasoning by sampling in amortized latent space. Specifically, this approach, Amortized Variational Inference (AVI) yields token- level learning through optimizing the Evidence Lower Bound (ELBO), which encourages diverse trajectories and provides a principled way to draw samples from the target posterior distribution.
This bayesian approach eliminates the reliance on the critic model as in BoN-related approach during rationale-solution searching at inference time.

**Questions:**

See Above.

**Ethical Concerns:**

["NO or VERY MINOR ethics concerns only"]

**Final Justification:**

After reviewing the author's rebuttal, I decide to maintain my score which still holds an up-word rating about this paper with accept.

**Limitations:**

yes

**Quality:**

3

**Strengths And Weaknesses:**

Strengths:

1. This paper proposes a novel training framework RGFN, achieved by the reward approximation and reference-guided policy exploration and a novel bayesian Inference over latent rationals approach.
2. Experimental results over 4 multimodal math reasoning benchmark demonstrate its effectiveness.
3. The approach is not only effectiveness but also efficient and get rid of extra critic model during inference searching.

Weakness:
1. Although the model is compared with SFT and GRPO results. Results on popular inference scaling approaches such as BofN etc. should also be compared with?
2. Is this approach the first to adopt Amortizing Variational Inference method, more comparison with related work is needed. Also comparison with existing latent space reasoning approaches?

---

> ### Author Rebuttal · Authors · 2025-07-31
>
> We thank the reviewer for their thoughtful comments and constructive suggestions. Below, we provide detailed responses to each point.
> > **`W3-1`**:  Although the model is compared with SFT and GRPO results. Results on popular inference scaling approaches such as BofN etc. should also be compared with?
>
> We appreciate the valuable suggestion. We provide comparison results of Best-of-N (BofN) approach below. Specifically, during inference, we sample N rational-answer pairs using LaCoT. For each pair, we compute the length-normalized log-likelihood of the generated answer as reward, and then select the final answer corresponding to the highest reward. To ensure a fair comparison, we do not use any external reward model.
>
> | Method            | MathVerse | MathVista | MMMU  | MMVet |
> |------------------|:---------:|:---------:|:-----:|:-----:|
> | 3B w/ BofN        |   21.2    |   57.1    | 44.7  | 67.1  |
> | 3B w/ BiN (ours)  | **40.0**  | **63.2**  | **48.8** | **69.6** |
> | 7B w/ BofN        |   26.5    |   62.2    | 47.3  | 71.2  |
> | 7B w/ BiN (ours)  | **39.7**  | **68.4**  | **54.9** | **74.2** |
>
>
> We set number of candidates N={5, 10} for BoN and BiN, and we report the highest score of each method.
>
> > **`W3-2`**: Is this approach the first to adopt Amortizing Variational Inference method, more comparison with related work is needed. Also comparison with existing latent space reasoning approaches?
>
>
> To the best of our knowledge, our work is the first to apply amortized variational inference to latent visual reasoning with a long CoT chain, where reasoning steps are learned and inferred in a latent space conditioned on both visual and textual input.
>
> Previous works ([1] and [2]) focus solely on the textual domain, applying these methods for latent visual CoT is nontrivial due to long and structured multi-step reasoning.
>
> [1] Hao, S., Sukhbaatar, S., Su, D., Li, X., Hu, Z., Weston, J.E., & Tian, Y. (2024). Training Large Language Models to Reason in a Continuous Latent Space. _ArXiv, abs/2412.06769_.
>
> [2] Geiping, J., McLeish, S., Jain, N., Kirchenbauer, J., Singh, S., Bartoldson, B.R., Kailkhura, B., Bhatele, A., & Goldstein, T. (2025). Scaling up Test-Time Compute with Latent Reasoning: A Recurrent Depth Approach. _ArXiv, abs/2502.05171_.

---

> ### Comment · Reviewer_tynq · 2025-08-05
>
> Thank the authors for their rebuttal, I keep my positive score about this work.

---

> > ### Author Response · Authors · 2025-08-05
> >
> > We are sincerely grateful for your continued support for the acceptance of this paper. Following your suggestions, we will incorporate our responses to your points into the revised version of the paper.
> >
> > Thank you again for your time and support!

---

### Official Review · Reviewer_fLHH · 2025-07-03

**Clarity:** 4
**Significance:** 4
**Originality:** 3
**Rating:** 5
**Confidence:** 2

**Summary:**

This paper proposes a method for improving chain-of-thought reasoning in vision-language models. The authors reformulate reasoning as a latent variable sampling problem and propose a method based on amortized variational inference using GFlowNets. In addition, the method includes a Bayesian inference method (BiN) for selecting optimal rationales at inference time. The proposed method show SoTA performances across several visual reasoning benchmarks.

**Questions:**

- How sensitive is the performance to the choice of lambda for approximating reward?
- What are training and inference times compared to baselines (SFT, GRPO)?
- How does the proposed reference-guided filtering compare to other exploration strategies like epsilon-greedy or entropy-based exploration?

**Ethical Concerns:**

["NO or VERY MINOR ethics concerns only"]

**Final Justification:**

The rebuttal considered additional evaluation datasets from different domains and addressed the reviewer's concern about inference costs. This reviewer will keep the original positive rating of this paper.

**Limitations:**

yes

**Paper Formatting Concerns:**

None.

**Quality:**

4

**Strengths And Weaknesses:**

**Strengths**:
- The authors propose a novel theoretical framework that reformulates reasoning as a latent variable sampling problem, and using amortized variational inference to improve upon existing methods.
- The proposed method shows strong improvements over prior methods.

**Weaknesses**:
- The proposed method is evaluated only on mathematical reasoning tasks. The generalizability to other domains remains unclear.
- The proposed method requires sampling multiple rationales at inference time, which is computationally expensive. The paper can benefit from a detailed analysis of computational costs compared to baselines.

---

> ### Author Rebuttal · Authors · 2025-07-31
>
> We thank the reviewer for their thoughtful comments and constructive suggestions. Below, we provide detailed responses to each point.
>
> > **`W2-1`**: The proposed method is evaluated only on mathematical reasoning tasks. The generalizability to other domains remains unclear.
>
> We appreciate the suggestion. We initially followed the R1-OneVision baseline to evaluate LVLM reasoning performance on mathematical benchmarks. To further assess the generalizability of **LaCoT** beyond mathematical reasoning, we conducted additional experiments on three diverse visual understanding benchmarks: **MMMU-pro**, **MMVet**, and **MME**. These benchmarks cover a broad range of tasks, including visual commonsense, fine-grained recognition, and multi-choice QA.
>
> | Method                | MMMU-pro | MMVet | MME  |
> |----------------------|----------|-------|------|
> | InternVL2-4B         | -        | 55.7  | 2046 |
> | Qwen2.5-VL-3B        | 22.4     | 61.4  | 2134 |
> | **LaCoT-Qwen-3B**    | **28.9** | **69.6** | **2208** |
> | InternVL2-8B         | 25.4     | 60.0  | 2210 |
> | Qwen2.5-VL-7B        | 34.6     | 70.5  | 2333 |
> | R1-Onevision         | 28.2     | 71.1  | 1111 |
> | **LaCoT-Qwen-7B**    | **35.3** | **74.2** | **2372** |
>
> These results demonstrate that LaCoT consistently improves performance across different domains and model scales, suggesting strong generalization beyond mathematical reasoning tasks.
>
>
> > **`W2-2`**: The proposed method requires sampling multiple rationales at inference time, which is computationally expensive. The paper can benefit from a detailed analysis of computational costs compared to baselines.
>
> We agree that rationale sampling introduces additional inference cost, and we address this by using mini-batching (with batch size k=5) to generate N rationales in N/k forward passes. Below, we report the average per-sample inference time (reasoning + answering) and corresponding performance of different reasoning-LVLM on MathVista and MathVerse:
>
> | #Rationales (N)        | 1    | 5     | 10    | MathVista | MathVerse |
> |------------------------|------|-------|-------|-----------|-----------|
> | LLaVA-CoT-11B          | -    | 340s  | 830s  | 52.5      | 22.6      |
> | R1-OneVision-7B        | 32s  | -     | -     | 64.1      | 37.8      |
> | **LaCoT-Qwen-7B**    | -    | 30s   | 65s   | **68.4**  | **39.7**  |
>
>
> Where LLaVA-CoT-11B utilizes stage-wise beam search [1] at inference time.
>
> As shown in the table, LaCoT-Qwen-7B consistently achieves stronger performance, even with modest increases in inference time. Compared to other multi-rationale baselines, LaCoT strikes a favorable balance between computational cost and reasoning reliability, thereby improving both the trustworthiness of rationales and the accuracy of final answers.
>
>
>
>
> > **`Q2-1`**:  How sensitive is the performance to the choice of lambda for approximating reward?
>
> We appreciate the valuable comment. We examined the sensitivity of performance to the choice of $\lambda$ in reward approximation by training with values in the range [8,32]. While the approximated rewards showed slight differences in early stages, they rapidly converged and became nearly identical after around 250 training steps, suggesting that LaCoT is relatively robust to the choice of $\lambda$ within a reasonable range.
>
> > **`Q2-2`**: What are training and inference times compared to baselines (SFT, GRPO)?
>
> We report the total training time of fine-tuning Qwen2.5-VL using different objective functions on 3k samples, all conducted on 8×A100 (80G) GPUs:
> | Method | Training Time | Inference Time/sample | MathVista  |
> |--------|---------------|-----------------------|------------|
> | SFT    | 1 hour        | 34s                   | 62.7       |
> | GRPO   | 64 hours      | 32s                   | 62.6       |
> | RGFN (ours)   | 90 hours      | 30s                   | 66.8 (N=5) |
>
>
> Where N indicates the number of explored rationals of BiN.
>
> To avoid out-of-memory (OOM) issues during GRPO and LaCoT fine-tuning, we employed DeepSpeed Stage 3 with gradient checkpointing enabled. While this setting substantially increased training time, it enabled on-policy exploration.
>
> These results demonstrate that while LaCoT incurs additional computational cost due to exploration and token-level reasoning supervision, it remains feasible for practical training pipelines and yields notable performance gains.
>
>
>
> > **`Q2-3`**: How does the proposed reference-guided filtering compare to other exploration strategies like epsilon-greedy or entropy-based exploration?
>
>
> We studied Epsilon-greedy in our early study, but we found that it does not scale well in training a reasoning LVLM. Specifically, the action space comprises the entire vocabulary (~50,000 tokens). In such a large space, most random actions are **semantically nonsensical** and low-reward, making exploration **inefficient** and often harmful. This results in **high-variance gradients** and **slow convergence**, leading to a catastrophic forgetting issue in our experiment.
>
> Entropy-based exploration methods such as PPO rely on optimizing policy gradients to maximize expected rewards. However, these methods often struggle to align the learned policy with a target distribution, especially in complex reasoning tasks. In contrast, our approach builds on the GFlowNet framework, which is designed to sample from a target distribution via trajectory-level credit assignment, making it better suited for tasks requiring diverse and calibrated generation [2].
>
> Our reference-guided filtering (RGFN) further enhances the stability of on-policy exploration by selectively retaining high-quality trajectories, ensuring both efficiency and robustness during training. This strategy offers a more principled and controllable alternative to stochastic exploration heuristics.
>
> [1] Xu, G., Jin, P., Li, H., Song, Y., Sun, L., & Yuan, L. LLaVA-CoT: Let Vision Language Models Reason Step-by-Step. ICCV 2025.
>
> [2] Hu, E.J., Jain, M., Elmoznino, E., Kaddar, Y., Lajoie, G., Bengio, Y., & Malkin, N. Amortizing intractable inference in large language models. ICLR 2024

---

### Official Review · Reviewer_B7EG · 2025-07-05

**Clarity:** 3
**Significance:** 2
**Originality:** 3
**Rating:** 4
**Confidence:** 3

**Summary:**

The paper introduces a new training algorithm for multimodal reasoning. Specifically, the paper reformulates reasoning in LMMs as a posterior inference problem, and proposes a new algorithm stemming from amortized variational inference. The paper then addresses some unique challenges in the multimodal domain, and proposes a sparse reward function for token-level learning signals. Additionally, the paper proposes a new inference time  scaling approach based on Bayesian inference over N latent rationales. Empirical results show effectiveness of the proposed algorithm against SFT and GRPO.

**Questions:**

1. Since LaCoT introduces an additional neural-based reward model, what would be the computational overhead compared to, e.g., GRPO?

**Ethical Concerns:**

["NO or VERY MINOR ethics concerns only"]

**Final Justification:**

After the authors add the new analysis results into next revision, and if there is room in the proceedings, the paper could be accepted.

**Limitations:**

Yes

**Quality:**

3

**Strengths And Weaknesses:**

Strengths

1. The main strength of the paper lies in the methodology part, which introduces a different view on how to learn multimodal reasoning models. The idea offers an interesting alternative beyond maximum likelihood (SFT) and outcome-supervised policy gradient (e.g., GRPO).

2. The description of the method is detailed and technically solid.

3. The additional exploration on Bayesian inference of the newly proposed method offers a valuable justification that LaCoT can also do effective inference-time scaling, which is one of the key benefits of current token-based explicit CoT models.

Weakness
1. The main concern lies in the experimental section which I find is a bit thin.

2. I wonder if the comparison with GRPO in Table 1 is fair, as LaCoT uses a different training set as R1-OneVision.

3. The performance of GRPO in Table 2 is a bit strange, as is contradictory with most recent papers observation. My empirical experience also indicates that GRPO works better than SFT especially in terms of generalization. The details of this ablation is not provided and no analysis on why this abnormal behavior happens.

---

> ### Author Rebuttal · Authors · 2025-07-31
>
> We sincerely thank the reviewer for their thoughtful feedback and constructive suggestions. Please find our detailed responses to each point below.
> > **`W1-1`**: The main concern lies in the experimental section which I find is a bit thin.
>
> We appreciate the suggestion. We conduct additional experiments on widely used benchmarks, including **MMMU-pro**, **MMVet**, and **MME**, where MMMU-Pro is a more robust version of MMMU, designed to assess LVLMs' understanding and reasoning capabilities more rigorously. These benchmarks cover a broad range of tasks, including visual commonsense, fine-grained recognition, and multi-choice QA.
>
>
>
> | Method                | MMMU-pro | MMVet | MME  |
> |----------------------|----------|-------|------|
> | InternVL2-4B         | -        | 55.7  | 2046 |
> | Qwen2.5-VL-3B        | 22.4     | 61.4  | 2134 |
> | **LaCoT-Qwen-3B**    | **28.9** | **69.6** | **2208** |
> | InternVL2-8B         | 25.4     | 60.0  | 2210 |
> | Qwen2.5-VL-7B        | 34.6     | 70.5  | 2333 |
> | R1-Onevision         | 28.2     | 71.1  | 1111 |
> | **LaCoT-Qwen-7B**    | **35.3** | **74.2** | **2372** |
>
>
>
> These results demonstrate that LaCoT consistently improves performance across different domains and model scales, suggesting strong generalization beyond mathematical reasoning tasks.
>
>
> > **`W1-2`**:   I wonder if the comparison with GRPO in Table 1 is fair, as LaCoT uses a different training set as R1-OneVision.
>
>
> Our SFT data is comparable with R1-OneVision. However, in the RL stage—which we consider the most critical for performance—R1-OneVision (GRPO) utilizes 10k newly collected samples. In contrast, LaCoT reuses only 3k samples from the SFT stage without additional data collection, which highlights its efficiency and generalization.
>
> > **`W1-3`**: The performance of GRPO in Table 2 is a bit strange, as is contradictory with most recent papers observation. My empirical experience also indicates that GRPO works better than SFT especially in terms of generalization. The details of this ablation is not provided and no analysis on why this abnormal behavior happens.
>
> In Table 2, we compare the effectiveness of different training algorithms on 3k samples, using consistent hyperparameters across all methods (e.g., same batch size, base model, and learning rate). For GRPO and RGFN (our approach), we set the exploration number to 6 and the sequence length to 700. While GRPO performs slightly worse than SFT, we think this is due to GRPO’s higher sensitivity to hyperparameters and its reliance on larger training data for stable generalization. This is also reflected in our previous experiments (R1-OneVision applies 10k training data), where its performance improves with more data.
>
> > **`Q1-1`**: Since LaCoT introduces an additional neural-based reward model, what would be the computational overhead compared to, e.g., GRPO?
>
> LaCoT introduces no additional GPU memory usage or data preprocessing overhead compared to GRPO during training. However, due to the token-level reward approximation, it does incur a higher computational cost in terms of runtime. Under identical experimental settings, LaCoT requires approximately **90 hours** of total training time, compared to **64 hours** for GRPO.

---

> > ### Comment · Reviewer_B7EG · 2025-08-05
> >
> > I've read the rebuttal and still think the paper has room to improve, so I would like to keep my score.

---

> > > ### Author Response · Authors · 2025-08-05
> > >
> > > Thank you for taking the time to read our rebuttal. We appreciate your feedback and we will carefully consider your suggestions to further improve the clarity and quality of the paper in the revisions.

---

### Note · Authors · 2025-08-14

We sincerely thank all reviewers for their constructive comments and suggestions. We have made every effort to address all concerns raised, and we are deeply grateful for the positive evaluations provided by all reviewers.

---

### Decision · Program_Chairs · 2025-09-17

**Decision:**

Accept (poster)

**Comment:**

This paper proposes a variational learning algorithm for reasoning in vision-language models in which chains-of-thought are treated as a latent variable, following prior work in latent CoT for text tasks. Performance is compared against SFT and GRPO on visual reasoning tasks including for math/abstract reasoning and visual commonsense. This appears to be an elegant work and I suggest updating the paper with the feedback from all reviewers, but especially AnFV, in mind.